# Feeding Spray-Dried Porcine Plasma to Pigs Improves the Protection Afforded by the African Swine Fever Virus (ASFV) BA71∆CD2 Vaccine Prototype against Experimental Challenge with the Pandemic ASFV—Study 2

**DOI:** 10.3390/vaccines11040825

**Published:** 2023-04-10

**Authors:** Joan Pujols, Elena Blázquez, Joaquim Segalés, Fernando Rodríguez, Chia-Yu Chang, Jordi Argilaguet, Laia Bosch-Camós, Rosa Rosell, Lola Pailler-García, Boris Gavrilov, Joy Campbell, Javier Polo

**Affiliations:** 1IRTA, Centre de Recerca en Sanitat Animal (CReSA), 08193 Barcelona, Spain; 2Unitat Mixta d’Investigació IRTA-UAB en Sanitat Animal, Centre de Recerca en Sanitat Animal (CReSA), Campus de la Universitat Autònoma de Barcelona (UAB), 08193 Barcelona, Spain; 3WOAH Collaborating Centre for Emerging and Re-Emerging Pig Diseases in Europe, IRTA-CReSA, 08193 Barcelona, Spain; 4APC Europe, S.L., 08403 Granollers, Spain; 5Departament de Sanitat i Anatomia Animals, Facultat de Veterinària, Campus de la Universitat Autònoma de Barcelona (UAB), 08193 Barcelona, Spain; 6Departament d’Acció Climàtica, Alimentació i Agenda Rural, Generalitat de Catalunya, 08007 Barcelona, Spain; 7Biologics Development, Huvepharma, 3A Nikolay Haytov Street, 1113 Sofia, Bulgaria; 8APC, LLC, Ankeny, IA 50021, USA

**Keywords:** African swine fever, ASFV, spray-dried porcine plasma, vaccine, challenge, nutritional intervention

## Abstract

This study aimed to evaluate the effects of feeding spray-dried porcine plasma (SDPP) on the protection afforded by the BA71∆CD2 African swine fever virus (ASFV) vaccine prototype. Two groups of pigs acclimated to diets without or with 8% SDPP were intranasally inoculated with 10^5^ plaque-forming units (PFU) of live attenuated ASFV strain BA71∆CD2 and, three weeks later, left in direct contact with pigs infected with the pandemic Georgia 2007/01 ASFV strain. During the post-exposure (pe) period, 2/6 from the conventional diet group showed a transient peak rectal temperature >40.5 °C before day 20 pe, and some tissue samples collected at 20 d pe from 5/6 were PCR+ for ASFV, albeit showing Ct values much higher than Trojan pigs. Interestingly, the SDPP group did not show fever, neither PCR+ in blood nor rectal swab at any time pe, and none of the postmortem collected tissue samples were PCR+ for ASFV. Differential serum cytokine profiles among groups at vaccination, and a higher number of ASFV-specific IFNϒ-secreting T cells in pigs fed with SDPP soon after the Georgia 2007/01 encounter, confirmed the relevance of Th1-like responses in ASF protection. We believe that our result shows that nutritional interventions might contribute to improving future ASF vaccination strategies.

## 1. Introduction

African swine fever virus (ASFV) is a large, enveloped, double-stranded DNA virus. As the only member of the *Asfarviridae* family [1], it can infect domestic pigs and wild boars of all ages causing African swine fever (ASF). ASF is a notifiable disease to the World Organization for Animal Health (WOAH, formerly OIE) and is the number one threat to the global swine industry and a major limitation to global trading [2]. Since 2007, a virulent ASFV strain (genotype II) has emerged, causing mortality rates up to 100%, and has spread in Europe, China, South-East Asia, and more recently, in the Dominican Republic and Haiti, where millions of animals have succumbed to the disease [3].

Currently, there is no commercial vaccine available to fight the ASF pandemic at a global level. The lack of efficacy of inactivated vaccines and the poor protection afforded so far with recombinant vaccines based on ASFV-specific antigens left live attenuated viruses (LAVs) as the short–medium term choice to develop ASFV vaccines [4,5,6]. The recent launching of the first commercial vaccine against ASFV in Vietnam was based on a recombinant deletion mutant lacking the I177L gene from the Georgia 2007 ASFV isolate [7] and reflects the high expectation that this technology has opened in the field. Unfortunately, vaccination in some regions of Vietnam was suspended due to unexpected pig deaths (https://www.reuters.com/world/asia-pacific/vietnam-suspends-african-swine-fever-vaccine-after-pigdeaths-2022-08-24/, accessed on 24 January 2023). This was most probably due to defects in vaccination implementation, reopening biosafety concerns about the use of ASF LAVs in the field [8].

Together with the necessary implementation of standardized protocols for registration and approval of ASFV vaccines, we should continue investing efforts to better understand the mechanisms involved in protection against ASF, aiming to develop the safest and most efficient preventive and therapeutic strategies. Both antibodies [9] and CD8+ cells [10] play important roles in protection, together with an appropriate innate immune response [11]. Recent work performed, among others, in our own laboratory has confirmed the key importance of Th1-like responses and specific cytotoxic T lymphocytes (CTLs) in protecting against ASF, independently of working with subunit experimental vaccines [12,13] or with BA71∆CD2, a cross-protective recombinant live attenuated virus [14].

In parallel to the current efforts to develop efficient vaccines against ASF, we and others have also demonstrated that the pig’s immune status and/or their microbiota significantly influence the disease outcome [15,16,17]. In the present study, we aimed to evaluate the effects of feeding pigs with spray-dried plasma (SDP) on the protection afforded by the BA71∆CD2 vaccine prototype. SDP derived from porcine (SDPP) or bovine (SDBP) origin are dry functional ingredients that are extensively used in pig starter diets and consistently improve performance, feed efficiency, and animal survival, especially under stressful conditions such as pathogen challenge [18]. SDP contains a diverse mixture of many functional compounds such as immunoglobulins, albumin, growth factors, biologically active peptides, transferrin, amino acids, and other molecules that have biological activity independent of their nutritional value. Although the modes of action of SDP are not completely known, it has been shown to modulate the efficiency of the immune system [19]. Based on the demonstrated capability of SDPP feeding to accelerate the induction of specific Th1-like responses and to delay experimental ASFV transmission and disease progression [20] (back-to-back submitted manuscript), the objective of this study was to evaluate the effects of feeding SDPP on the protection afforded by the BA71∆CD2 vaccine prototype. 

## 2. Materials and Methods

### 2.1. Clinical Monitoring

The clinical state of the animals and the end-point criteria were evaluated by scoring the ASF-compatible clinical signs following a previously reported guide [21] with slight modifications. A score from 0 to 5 according to severity was applied as follows: 0: no clinical signs; 1: mild pyrexia (39.6–40.0 °C); 2: mild pyrexia (39.6–40.0 °C) and mild clinical signs (skin, digestive); 3: moderate pyrexia (40.0–40.5 °C) and mild–moderate clinical signs (distal ear spots, mild limp, lying down, but remaining alert); 4: moderate–high pyrexia (40.5–41 °C) and moderate clinical signs (remains dormant, only stands up when touched, hesitant step, subcutaneous bleeding <10%, diarrhea, mild tremors); and 5: pyrexia higher than 41 °C and moderate–severe clinical signs (generalized subcutaneous bleeding, ataxia, spasticity, clouding, prostration, bloody diarrhea). 

### 2.2. Study Design

The study was approved by the Committee of Ethics and Welfare “Comitè d’Experimentació Animal de la Generalitat de Catalunya” with the protocol approval number CEA-OH/11387/1. For this study, 24 4-week-old Landrace x Large White male pigs were used. Pigs were randomly divided into two separate groups at the IRTA-Monells animal facility and acclimated to their assigned diet (see Table 1, as described by Blazquez et al. [20]).

Thus, sixteen animals were fed a conventional diet with 10.09% soy protein concentrate, and eight pigs were initially fed a diet supplemented with 8% SDPP (AP920 produced by APC Europe S.L.U.-Granollers, Spain), replacing soy protein. Then, 14 days later, animals were moved to the IRTA-CReSA BSL-3 animal facility and were distributed in two separate rooms (Rooms 1 and 2) divided in half (left and right) with fences, maintaining the same diet (8 fed with SDPP-supplemented diet and 16 with the conventional diet) during 10 additional acclimation days, as described in Figure 1. The rooms contain slatted floors, and the environmental conditions for both rooms were set at 22 ± 2 °C and relative humidity of 60 ± 5%. The air renewal was established to be 12 times/h. The feed was provided each morning between 7:30 and 9:30 a.m.

After acclimation, eight pigs from Room 1 (left pen), fed the conventional diet, and eight pigs from Room 2 (left pen), fed the SDPP diet, were intranasally immunized (1 mL per nostril) with a dose of 10^5^ PFU of BA71ΔCD2 ASFV. The rest of the pigs (eight), four located in the right half of Room 1 and four in the right half of Room 2, continued to be fed the conventional diet. Therefore, the donor animals (Trojans) for both groups were fed the same control diet.

Nineteen days after vaccination (d19pv), each of the non-immunized animals (two groups of four pigs fed with the conventional diet) was intramuscularly infected with 1 mL of a lethal dose (10^3^ GECs/mL) of the pandemic Georgia 2007/1 ASFV strain [22]. Finally, two days later, the fences were removed, allowing the direct contact of vaccinated pigs with the “Trojan” pigs previously infected with Georgia 2007/01 (day 0 post-exposure, d0pe), with a final 1:2 ratio of Trojan to vaccinated pigs. This proportion was considered optimal for the transmission of the virus by direct contact [14]. All Trojan pigs had to be sacrificed between 3 and 7 days post-infection. The study ended at d20pe, 41 days after initiating the vaccination (Figure 1).

Throughout the experimental period, pigs were fed ad libitum and observed for clinical signs and rectal temperature daily. Blood samples (10 mL tubes with EDTA) and nasal and rectal swabs were taken at d0pv (before intranasal vaccination), d7pv, d14pv, and d21pv (d0pe, the first day the Trojan and vaccinated pigs were in contact), d4pe, d7pe, d14pe, and d20pe (d41pv). At necropsy, lesions were registered, and samples of spleen, tonsil, and gastro-hepatic, submaxillary, and retropharyngeal lymph nodes were frozen and kept at −75 °C until use. Once thawed, all samples were simultaneously analyzed with real-time PCR (qPCR) to detect ASFV virus loads, as described below [23]. 

ELISPOT analysis was conducted using fresh peripheral blood mononuclear cells (PBMCs) from blood obtained with EDTA at d21pv (d0pe) and at d9pe and d20pe.

### 2.3. Laboratory Analyses

DNA extraction was performed using an Indimag Pathogen Kit (Indical Biosciences, Leipzig, Germany). Viremia was determined with qPCR analysis using the primers described by Fernández–Pinero et al. [23] and the probe ASF-VP72P1 described in the current OIE ASF chapter (Terrestrial Manual OIE, Section 3.9, Chapter 3.9.1 African Swine Fever Virus pages 1–18) with the following modification in the thermoprofile made by the Spanish National Reference Laboratory for ASF: 10 min at 95 °C, 5 cycles 1 min at 95 °C + 30 s at 60 °C, 40 cycles 10 min at 95 °C + 30 s at 60 °C with fluorescence acquisition in the FAM channel at the end of each PCR cycle. According to these amplification settings, results were considered positive when Ct ≤ 30, inconclusive when Ct was between 30 to 35, and negative when Ct > 35. 

The differential detection of BA71ΔCD2 was performed in blood samples after exposure using a recently described probe-based SYBR Green qPCR (Applied Biosystems, Waltham, MA, USA, Path-ID qPCR Master Mix), targeting the LacI reporter gene, only present in the genome of the BA71ΔCD2 vaccine virus [14]. The detection limit of this qPCR was 20 copies/reaction, with a Ct value of 34.93, standardized using serial dilutions of a plasmid encoding the LacI gene.

Seroconversion was determined with ELISA (INgezim PPA COMPAC, INGENASA; Madrid, Spain). Nasal and rectal swabs were analyzed for the presence of the ASFV viral genome using the procedures previously mentioned. Once the nasal and rectal swabs arrived at the laboratory, the end of the swab was cut and placed in a tube with 1 mL of PBS. Tubes were stored at −75 °C until DNA extraction and analysis with qPCR.

For each tissue sample, 0.1 g of tissue was diluted 1:10 and homogenized using sterile PBS and TyssueLyser II (Qiagen, Hilden, Germany); DNA extraction and qPCR were performed as detailed above.

Plasma samples obtained from pigs were stored at −75 °C until use. Once thawed, levels of IFNα, IFNγ, IL-1β, IL-10, IL-12/IL-23p40, IL-4, IL-6, IL-8, and TNFα in plasma were quantified using the Luminex xMAP technology following the manufacturer’s instructions (ProcartaPlex Porcine Cytokine & Chemokine Panel 1; ThermoFisher Scientific, Waltham, MA, USA). The concentrations of each cytokine were calculated using xPONENT software (Luminex, Austin, TX, USA). Levels of TGFβ and IL-17α were quantified with ELISA (KingFisher Biotech, Saint Paul, MN, USA, [DIY0730S-003] and Invitrogen [CHC1683Kit], Waltham, MA, USA, respectively), following the manufacturer’s instructions.

PBMCs were purified from EDTA blood samples by density–gradient centrifugation with Histopaque 1077 (Sigma-Aldrich, St. Louis, MO, USA). To quantify by ELISPOT assay the number of IFNγ secreting cells, fresh PBMC were stimulated for 16 h with BA71∆CD2 (vaccine prototype) and/or Georgia 2007/01 (pandemic challenge virus) at a multiplicity of infection (MOI) of 0.2. Commercial mAbs (Porcine IFNγ P2G10 and biotin P2C11, BD Biosciences Pharmingen, San Diego, CA, USA) at 5 µg/mL were used, as previously described [24] (Díaz and Mateu 2005). Plates were revealed using HRP-conjugated Streptavidin (Life Technologies, South San Francisco, CA, USA) and TMB substrate (MABTECH, Stockholm, Sweden), and spots were counted under a magnifying glass.

### 2.4. Statistical Analysis 

Data were analyzed as a completely randomized design using the GLM procedures of SAS (SAS Inst., Inc., Cary, NC, USA). An analysis of variance was conducted to detect differences among treatments. The independent variable was treatment. Dependent variables were body temperatures, blood, nasal and rectal swab, and tissue Ct values. The LSMEANS procedure was used to calculate the mean values by treatment. If treatment effects were detected, least squares means were separated using the PDIFF option in SAS. For cytokine statistical analysis, a linear mixed-effect model was constructed for each cytokine, with animal and replicate as fixed effects and treatment and day of study as random effects. In order to contrast both treatments at each time point, a post-hoc analysis was performed. The pig was considered the experimental unit. Means are considered significantly different if *p* < 0.05, while trends are reported as *p* = 0.05 to 0.10.

## 3. Results

In the group fed the conventional diet, one animal died before starting the vaccination period. Therefore, this group started with 7 animals instead of 8. In addition, another animal (#396) died in this group on d31pv due to acute meningitis. In the SDPP group, 1 animal (#391) was euthanized on d21pv to balance both groups to 7 animals during the exposure period. This animal (#391) was randomly selected between animals within the corresponding pen, excluding those showing the lowest and the highest rectal temperature. Another animal (#376) died on d35pv due to intestinal prolapse. Both groups finished at d41pv with 6 pigs. 

All pigs intranasally vaccinated with 10^5^ PFU of BA71∆CD2 survived the direct-contact challenge with pigs infected with Georgia 2007/01 independently of their assigned diet. However, differences were observed between treatment groups, including clinical signs and viral load in blood, excretions, and tissues after exposure to the ASFV-infected Trojan pigs. Interestingly, no fever was recorded at any time after Georgia 2007/01 challenge in any of the pigs from the SDPP group (Figure 2A; Appendix A). 

A proportion of pigs vaccinated with 10^5^ PFU intranasally and fed the conventional diet (C3 and C4) showed a peak of fever starting at d14pe (Figure 2B), with no other signs of ASF infection observed. 

In addition to elevated rectal temperature, pigs #C3 and #C4 were the only ones exhibiting significant virus loads in both their blood and nasal swabs at the end of the experiment (Figure 3B; Appendix A).

Pigs fed the SDPP-containing diet did not exhibit fever or become viremic, and virus was not detected in rectal swabs at any time post-Georgia 2007/01 exposure (Figure 3A; Appendix A). 

In addition, the vaccine provided sterilizing protection in pigs fed the SDPP-containing diet, as no virus was detected in tissue samples at the end of the study (Figure 4A). In contrast, virus was present in most of the pigs from the conventional diet group in at least one organ at d20pe (Figure 4B; Appendix A).

Without exception, the average Ct values found in vaccinated pigs, both in fluids during the exposure period and in postmortem tissues, were much higher than those found in Trojan pigs succumbing to the ASFV challenge (see bottom panels in Figure 3 and Figure 4; Appendix A) confirming solid protection afforded by BA71∆CD2. Finally, in all cases, the only virus detectable after Georgia 2007/01 exposure was the pandemic virus, with no detectable traces of the BA71∆CD2 vaccine prototype (Appendix A).

No treatment differences were observed (*p* > 0.1) in the kinetics of induction of ASFV-specific IgG and IgA in sera between either treatment group of pigs before and after the challenge, except for d20pe (d40pv) in which the average values for IgG and IgA in conventional pigs was higher (*p* < 0.039 for IgG and *p* < 0.092 for IgA) (Figure 5A,B; Appendix A). 

Similarly, treatment differences were not found (*p* > 0.1) between the number of ASFV specific IFNγ-secreting T cells present at d0pe and d20pe (Appendix A). However, by d9pe, the number of specific T cells secreting IFNγ upon ASFV stimulation was numerically higher for the vaccine virus (BA71∆CD2), and a trend to be higher (*p* = 0.07) for the Georgia 2007/01 pandemic ASFV strain in pigs fed the SDPP diet than the conventional diet (Figure 5C). 

Finally, the immunomodulatory influence of the SDPP-containing diet was evident after 24 days of feeding, even before starting the vaccination. Indeed, pigs fed with SDPP showed a modest but detectable amount in serum of both pro-inflammatory (IL-8 and TNFα) and anti-inflammatory cytokines (IL-4) at vaccination time (Figure 6). 

Interestingly, the levels of the pro-inflammatory cytokine IL-17 before vaccination were lower in the SDPP group than in pigs fed with a conventional diet, and, as expected, the levels of cytokines in serum varied along the experiment. Furthermore, the Th1 cytokine IL-12/IL-23p40 increased in the SDPP group at 7 dpv (Figure 6), coinciding with the elevated levels of ASFV-specific T cells observed with ELISPOT (Figure 5C). Levels of the two cytokines IFNγ and TNFα, also recently identified as markers of vaccine-induced ASFV-specific Th1 response [14], showed a peak at 4 dpe significantly higher in animals fed with SDPP. Despite TGFβ seeming to show some significant differences, they mostly corresponded to the outlier value observed for one pig in the conventional group (#C4), coinciding with an animal showing fever and low Ct values. Finally, IL-8, TNFα, and IL-10 peaked at the end of the study in the group of pigs fed the conventional diet, coinciding with the detection of ASFV replication and fever (Appendix A).

## 4. Discussion

Due to the wide expansion of ASFV around the world, leading to a negative impact on pig health as well as huge economic consequences for pig producers, research on new treatments and vaccines that could help to mitigate the negative repercussions related to this virus has intensified in recent years. Because inactivated and subunit vaccine formulations have failed to protect pigs against the pandemic ASFV, research efforts have focused on either natural or recombinant LAVs as the only short–medium-term strategy to obtain highly efficient ASF vaccines [4,25]. Several recombinant LAV candidates have been described so far in the literature, capable of inducing solid protection against experimental challenges with the pandemic genotype II ASFV strains [5,6].

The detection of illegally introduced genotypes I and II attenuated ASFV vaccines in Chinese pig farms [2,26] confirms the need for caution when delivering ASF LAVs to the field without the appropriate supervision from regulatory agencies. This led to renewed efforts to standardize protocols for the registration and approval of the most efficient and safest vaccines [27]. At the same time, it is important to continue searching for methods to improve LAV prototypes and to develop efficient subunit vaccines for the future. 

In the present experiment, we extended previous work using BA71ΔCD2, a recombinant vaccine prototype capable of protecting in a dose-dependent manner against experimental challenges with homologous and heterologous viruses, including the genotype II pandemic virus [28,29]. For comparative studies, in the present work, an intranasal dose of 10^5^ PFU of BA71ΔCD2, one logarithm below the optimal dose previously reported [14], was administered to pigs fed a conventional diet with or without SDPP. As expected for the dose and route used, all pigs survived exposure to the Trojan pigs infected with Georgia 2007/01, independently of the dietary treatment. However, significant differences were observed between treatment groups. Tissues and fecal samples from all pigs consuming the SDPP-containing diet were negative for ASFV genome detection (at levels below our detection methods) at all sampling times following exposure to the Trojan pigs. In contrast with this apparent sterilizing protection, many pigs fed the conventional diet showed detectable virus in one or more samples post-exposure to the Trojan pigs. It is important to notice that the Georgia 2007/01 virus titers found in fluids and tissues of some vaccinated pigs fed the conventional diet were higher than expected, at least compared with those previously found using lower and higher vaccine doses than the one tested here [14]. We believe that this might be due to a sub-optimal health status of the animals from origin, which might negatively affect ASFV vaccination and transmission, as has been postulated in [14,17]. In fact, and as described in the results section, we had three ASF-unrelated deaths, one of them even before starting the vaccination despite treating them with antibiotics. Independently of this reality, the virus titers found in vaccinated pigs were always below those found in the Trojan pigs succumbing to Georgia 2007/01, confirming the solid protection afforded by the vaccine, even in adverse conditions. Therefore, the current data suggest that dietary SDPP improved vaccine efficiency and demonstrates that dietary SDPP could have a direct benefit for ASF vaccination. 

Treatment differences, except at the end of the study, were not observed in the serum levels of anti-ASFV IgG and IgA antibodies. Further studies are needed to better understand the kinetics of induced immunoglobulin isotypes as well as the local immune responses induced at the site of immunization and ASFV entry (nasal mucosa) and other mucosal tissues to distinguish any changes in the antibody induction due to feeding diets with SDPP [30,31].

Conversely to the antibody kinetics, an increase in specific IFNγ secretory T cells was observed in vaccinated pigs, detectable by d9pe to the Trojan pigs. The fact that virus-specific T cells recognize both the BA71∆CD2 vaccine and the pandemic virus confirms the induction of T cells capable of recognizing genotypes I and II, currently circulating in China [2]. These results and the corresponding peak of serum TNFα early after Georgia 2007/01 exposure of SDPP-fed pigs is consistent with the relevance of vaccine-induced IFNγ + TNFα+ polyfunctional memory Th1 cells in ASFV protection [14] and in the delayed transmission of ASFV in SDPP-fed pigs [20]. Additionally, in this line, the peak of the Th1 cytokine IL12 [32] in plasma at 7 dpv also suggests the SDPP-driven enhancement of the vaccine-induced Th1 response. Altogether, these results indicate that the addition of SDPP to diet somehow enhances the vaccine-induced ASFV-specific cellular responses. Further studies focused on mucosal immunity and its interplay with systemic immune responses will be required to better characterize the mechanisms behind this observation.

Of particular interest are the differences observed for several immune mediators in the serum of vaccinated pigs consuming the SDPP diet compared to that of pigs fed the conventional diet, even before immunization. Feeding SDPP resulted in elevated levels of pro- and anti-inflammatory cytokines in their plasma, confirming the tight regulation of the immune responses previously reported for SDPP-fed animals [33,34]. These results also confirm the relevance of the health and immune status of the animals in the responses observed after ASFV vaccination [15,16,17] and open new avenues for dietary intervention. 

Interestingly, the sequential increase in IL-17 (from d0pv to d21pv) in vaccinated pigs fed the conventional diet may indicate a tighter control of Th17 and T-regulatory cells favoring the optimal expansion of Th1-like responses [35]. In this regard, a recently published study associated the presence of regulatory T cells with the lack of long-term memory responses induced by ASF LAVs [36].

Dietary SDPP was shown to improve vaccine efficiency in a commercial trial where pigs were vaccinated with a commercial vaccination program [37]. Finally, the increase in IL-10, IL-8, and TNFα at the end of the study in pigs fed the conventional diet is consistent with the presence of Georgia 2007/01 at this time point in the animals from this group. Of course, the theoretical dysregulation of these cytokines is much lower than previously described at a late time post-infection in naïve pigs showing acute ASF with an uncontrolled cytokine storm due to the massive ASFV presence [38,39,40,41].

SDPP is a functional feed ingredient that has been demonstrated to systemically modulate the immune system. In some studies, the effects exerted by SDPP were characterized to promote both Th1-like response and cytokine induction. Díaz et al. [42] reported a significant reduction in interstitial pneumonia and faster virus clearance in pigs infected with porcine reproductive and respiratory syndrome virus and consuming an SDPP-containing diet. The authors concluded that pigs fed SDPP were able to mount a more robust immune response and were able to clear the virus more efficiently. The increase in cytokine expression (IFNγ and IL-1) in the lungs of pigs receiving SDPP in their feed points to a Th1 enhancement. Markowska-Daniel and Pejsak [43] reported that the administration of SDPP through both water and feed significantly increased, especially in smaller pigs, the percentage of CD8+ T cells, which are critical for protection against ASFV [10]. Thus, dietary SDPP may enhance the immune response of pigs vaccinated with BA71∆CD2, contributing to sterilizing immunity observed in the present experiment.

## 5. Conclusions

In summary, under the conditions of this exploratory study to prove the concept, the addition of SDPP in feed improved the ASFV vaccine prototype efficacy. Pigs fed the diet with SDPP showed lower virus load in nasal secretion and absence of virus in blood and feces after exposure to Trojan pigs infected with Georgia 2007/01 compared to those fed the conventional diet. Furthermore, no virus was detected in any organ tissue of the pigs fed the SDPP diet at the time of sacrifice (d20pe). This suggests that dietary SDPP can be used strategically as a health management tool to enhance vaccine efficiency. However, it is important to point out that the current work involved a limited number of animals under controlled conditions. These results should be verified and extended in further studies with a larger number of animals under field conditions.

## Figures and Tables

**Figure 1 vaccines-11-00825-f001:**
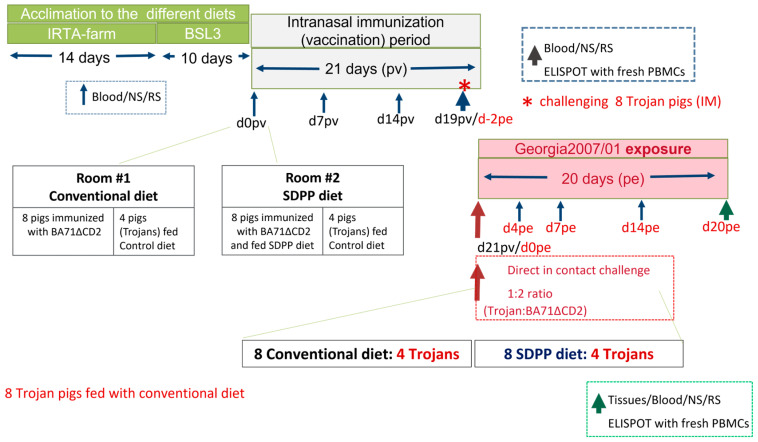
Schematic representation of the study design. Abbreviations: BSL3 = biosafety level 3; NS = Nasal swab; RS= rectal swab; pv = post-vaccination; pe = post-exposure; d = day; PBMCs = peripheral blood mononuclear cells.

**Figure 2 vaccines-11-00825-f002:**
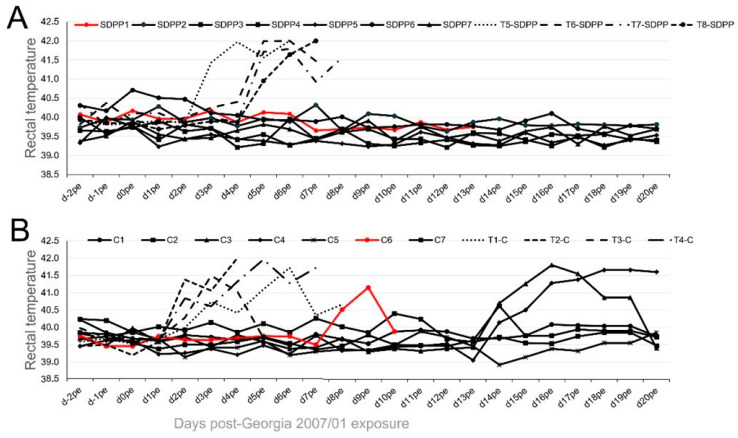
Rectal temperature recorded individually after Georgia 2007/01 exposure of pigs intranasally inoculated with 10^5^ PFU of BA71∆CD2. Data from pigs fed (**A**) the SDPP diet (SDPP1-7 pigs) or (**B**) the conventional diet (C1–C7 pigs). T1–T8 (dashed lines): Trojan pigs used for the direct contact challenge. In red color, animals that died before the end of the study.

**Figure 3 vaccines-11-00825-f003:**
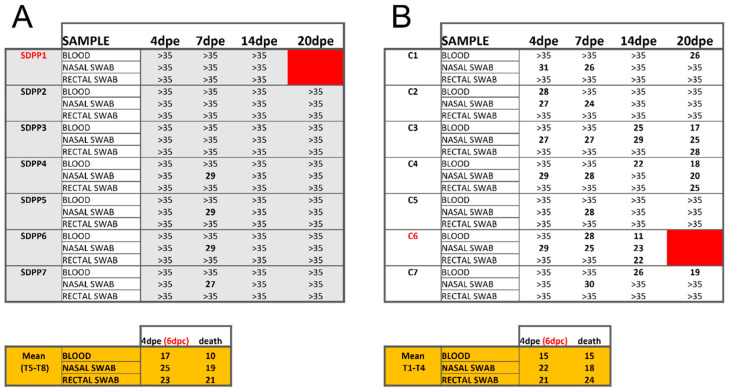
Ct values obtained by qPCR using blood, nasal and rectal swabs from individual animals at different times post-Georgia 2007/01 exposure (pe). Data from pigs fed (**A**) the SDPP diet (SDPP1-7 pigs) or (**B**) the conventional diet (C1–C7 pigs). The higher the Ct values, the lower the ASFV load, with Ct values > 35 being considered negative (cut-off of the technique). Tables in orange show the average Ct values obtained in samples from Trojan pigs. Animals in red color died before the end of the experiment.

**Figure 4 vaccines-11-00825-f004:**
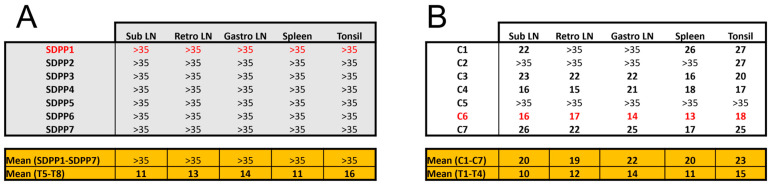
Ct values obtained by qPCR from different tissue samples of individual animals at the end of post-Georgia 2007/01 exposure (d20pe). Data from pigs (**A**) fed the SDPP diet (SDPP1-7 pigs) or (**B**) the conventional diet (C1–C7 pigs). The higher the Ct values, the lower the ASFV load, with Ct values > 35 being considered negative (cut-off of the technique). Tables in orange show the average Ct values obtained in samples from Trojan pigs. Animals in red color died before the end of the experiment. Abbreviations: Sub LN = Submaxillary lymph nodes; Retro LN = Retropharyngeal lymph nodes; Gastro LN = Gastro-hepatic lymph nodes.

**Figure 5 vaccines-11-00825-f005:**
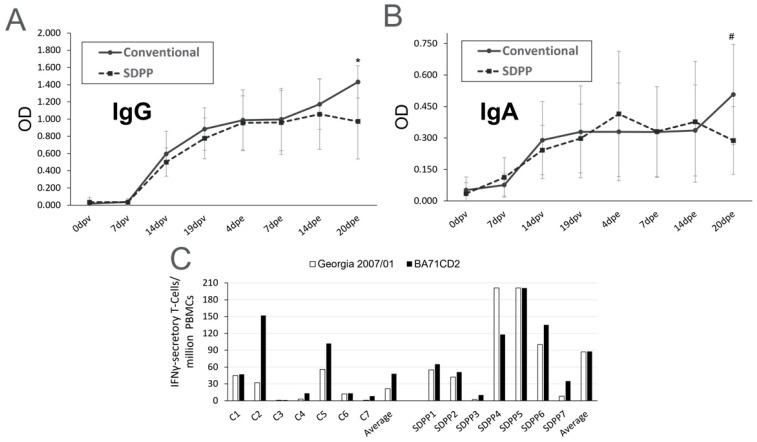
ASFV-specific immune responses analyzed in BA71∆CD2-vaccinated pigs before and after Georgia 2007/01 challenge. ASFV-IgG (**A**) and IgA (**B**) antibodies from sera quantified by ELISA. Average values for SDPP-fed pigs (SDPP1-7) are shown as dashed lines, while solid lines represent average values for pigs fed the conventional diet (C1–C7). Standard deviation values are shown. (**C**) Number of ASFV-specific IFNγ-secreting T cells found at d9pe by ELISPOT in PBMCs upon in vitro stimulation with either the pandemic Georgia 2007/01 (white boxes) or the BA71∆CD2 vaccine (black boxes). * = *p* < 0.05; # = *p* < 0.1.

**Figure 6 vaccines-11-00825-f006:**
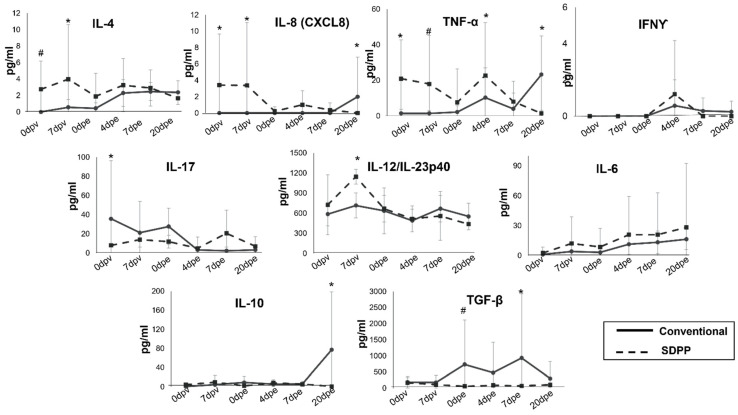
Serum concentrations of several cytokines quantified by Luminex after 24 days of diet acclimation (before immunization). * = *p* < 0.05; # = *p* < 0.1.

## Data Availability

All data from this study are provided in the manuscript and Appendix A.

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
