# Peer review of "Feeding Spray-Dried Porcine Plasma to Pigs Improves the Protection Afforded by the African Swine Fever Virus (ASFV) BA71∆CD2 Vaccine Prototype against Experimental Challenge with the Pandemic ASFV—Study 2"

_vaccines, 2023, doi:10.3390/vaccines11040825_

Round 1

Reviewer 1 Report

Following study 1, the authors further investegated the possible effect of SDPP diet on animals vaccinated with genetically modified live attenuated vaccine and exposed to trojan animals. The authors benefeted from the outcomes of the first study to conduct the 2nd trial. It was interesting to see that SDPP-fed and vaccinated  animals were protected against ASFV. However, this trial is missing the actual challenge of SDPP-fed and vaccinated animals by direct inoculation of ASFV into these animals.

1- line 187: why this particular animals was selected to be euthanized. what was the selection criteria?

2- Fig. 5C is not complete; the bars are cut from the top.

Author Response

Reviewer #1:

Following study 1, the authors further investigated the possible effect of SDPP diet on animals vaccinated with genetically modified live attenuated vaccine and exposed to trojan animals. The authors benefited from the outcomes of the first study to conduct the 2nd trial. It was interesting to see that SDPP-fed and vaccinated animals were protected against ASFV. However, this trial is missing the actual challenge of SDPP-fed and vaccinated animals by direct inoculation of ASFV into these animals.

We appreciate the statement from the reviewer #1. We really believe that the more solid protection afforded by the SDPP and vaccinated animals compared with that observed in conventional-fed and vaccinated animals deserves publication.

Understanding the concern of the reviewer, we have routinely confirmed that the “in contact ASFV challenge” is as reproducible as the direct inoculation pf ASFV model. This is not a challenge sporadically performed in the laboratory (for example in Bosch-Camós et al., 2022), but a 100% efficient challenge model, reliable if keeping the ratio of Trojans versus contact pigs. In fact, the experiments shown in the back-to-back submitted manuscript. Entitled “Feeding spray-dried porcine plasma to pigs reduces African swine fever virus load in infected pigs and delays virus transmission. Study 1.” confirmed once again the reliable nature of a model that could be translated to any other laboratory, always keeping the mentioned ratio. Moreover, we consider that this is a more natural model of infection compared to the one in which we inject the virus intramuscularly.

Ethical restrictions (mandatory by our authorities) made impossible to justify duplicating the number of groups and animals used at once, so future experiments will confirm the results observed by using alternative challenge models including intramuscular and intranasal ASFV inoculation with the Georgia07 strain. Despite this additional information will be extremely useful for the field, we believe that our data is solid enough to deserve publication.

1- line 187: why this particular animal was selected to be euthanized. what was the selection criteria?

This animal (#391) was randomly selected between animals within the corresponding pen, excluding those showing the lowest and the highest rectal temperature.

A sentence has been added to the new version of the manuscript. Line 202-203.

2- Fig. 5C is not complete; the bars are cut from the top.

Thank you for the comment and apologies for the mistake. The corrected version of the manuscript includes a new Fig. 5C.

Reviewer 2 Report

Pujol et al demonstrate the effect of spray dried porcine plasma on the immune responses after vaccination with an experimental live attenuated African swine fever virus when compared to conventional feed. The authors then contact infected the animals with donor pigs and found the group of animals fed SDPP did not become infected. Unfortunately the donor pigs used to infect the two groups of animals were fed on different diets and it appears this has impacted the transmission window meaning the two groups can't be directly compared. Based on the clinical scores in Fig 1 the SDPP pigs were in contact with at least one infected pig for 6 days whereas the conventionally fed pigs were in contact for seven. Crucially there were higher virus titres 4 dpe in all of the donor conventionally fed pigs compared to the SDPP. This means the infectious dose for the two experiments were different. As such the authors conclusions aren't supported by the data as the difference could be due to the health status of the SDPP fed animals (as the authors conclude) or the SDPP donor animals weren't sufficiently infectious. I note that this paper has a companion manuscript which may answer some of these questions and therefore I have selected Major Revision.

Other comments

Line 67 to 69. I don't understand the sentence, please clarify "These findings were obtained independently of working with sub-unit experimental vaccines".

Line 99. Study design doesn't include an ethical statement. It is surprising this study didn't include control groups of naive pigs to confirm transmission, I note that these may be described in the second paper. As per the ARRIVE guidelines power calculations should be included to confirm the study design, particularly if there is no control group. As this is a transmission experiment details of the cleaning regime and room conditions (temperature, humidity, air changes etc) should be included.

Line 231 to 233. Please include the data demonstrating that the virus was Georgia 2007/1 and not the vaccine.

Figure 5C. Figure seems to be truncated at 140 SFU. Any statistics should be based on the actual values.

Figure 6. Please report the actual P values for these comparisons, the variance in the data is so wide that I question whether the statistical differences are biologically significant? Were the data analysed using repeated measures ANOVA/Mixed Effect Model?

Line 361. Oura et al. compares pigs with and without CD8 cells, it is difficult to draw the conclusion stated by the authors about elevated CD8 T-cells. Takamatsu 2013 and Goatley 2022 shows elevated CD8 cells in vaccinated animals that are suffering from disease.

Author Response

Reviewer #2:

Pujol et al demonstrate the effect of spray dried porcine plasma on the immune responses after vaccination with an experimental live attenuated African swine fever virus when compared to conventional feed. The authors then contact infected the animals with donor pigs and found the group of animals fed SDPP did not become infected. Unfortunately, the donor pigs used to infect the two groups of animals were fed on different diets and it appears this has impacted the transmission window meaning the two groups can't be directly compared. Based on the clinical scores in Fig 1 the SDPP pigs were in contact with at least one infected pig for 6 days whereas the conventionally fed pigs were in contact for seven. Crucially there were higher virus titers 4 dpe in all of the donor conventionally fed pigs compared to the SDPP. This means the infectious dose for the two experiments were different. As such the authors conclusions aren't supported by the data as the difference could be due to the health status of the SDPP fed animals (as the authors conclude) or the SDPP donor animals weren't sufficiently infectious. I note that this paper has a companion manuscript which may answer some of these questions and therefore I have selected Major Revision.

We appreciate the comments from the Reviewer #2. However, we believe that there is a confusion in the interpretation, most probably due to the lack of clarity on our side.  The donor animals (Trojans) for both groups were fed with the same control diet, so there is not any difference on this regard. Agreeing with the reviewer that this might not read with the clarity that this crucial issue deserves we have rephrased the corresponding section of the manuscript section (2.2 of Materials and methods) and the Figure 1.   

Regarding second concern from the reviewer # 2, it might be also influenced by the lack of clarity mentioned before. We must keep in mind that the donor animals (Trojans) for both groups were fed with the same control diet, so there is not any difference on this regard.  More specifically, we would like to bring the attention over the figure 3 and supplementary table S2, showing that despite the slightly different kinetics on virus secretion observed, both groups of Trojans were apparently exposed and exerted to a similar virus pressure and, ultimately, even a more prolonged contact (and therefore exposure) was suffered by the SDPP-fed vaccinated and in contact animals. We really believe that the differences observed were not due to the challenge dose of ASFV but to the better immune status of the SDPP vaccinated pigs.

Other comments

Line 67 to 69. I don't understand the sentence, please clarify “These findings were obtained independently of working with sub-unit experimental vaccines.

We agree with the reviewer that the sentence was not clear enough. It has been modified to the following:

“Recent work performed among others in our own laboratory has confirmed the key importance of Th1-like responses and specific cytotoxic T lymphocytes (CTLs) in protection against ASF, independently of working with subunit experimental vaccines [12,13] or with BA71∆CD2, a cross-protective recombinant live attenuated virus [14]”.

Line 99. Study design doesn't include an ethical statement. It is surprising this study didn't include control groups of naive pigs to confirm transmission, I note that these may be described in the second paper. As per the ARRIVE guidelines power calculations should be included to confirm the study design, particularly if there is no control group. As this is a transmission experiment details of the cleaning regime and room conditions (temperature,humidity, air changes etc) should be included.

This study was approved by the committee of ethics and welfare “Comitè d’Experimentació Animal de la Generalitat de Catalunya” with the protocol approval number CEA-OH/11387/1 as appear in the “Institutional Review Board Sattement (lines 417-419).

Following the reviewer suggestion, we added this sentence in new line 100-101 of the revised manuscript: The study was approved by the committee of ethics and welfare “Comitè d’Experimentació Animal de la Generalitat de Catalunya” with the protocol approval number CEA-OH/11387/1.”.

As the reviewer #2 noted, this paper has a companion manuscript that confirmed once again the reliable nature of a model that could be easily translated to any other laboratory, always keeping the mentioned infected pig:in-contact pig ratio. Understanding the concern of the reviewer, we have routinely confirmed that the “in contact ASFV challenge” is as reproducible as the direct inoculation pf ASFV model. This is not a challenge sporadically performed in the laboratory (for example in Bosch-Camós et al., 2022), but a 100% efficient challenge model, reliable if keeping the ratio of Trojans versus contact pigs. In fact, the experiments shown in the back-to-back submitted manuscript entitled “Feeding spray-dried porcine plasma to pigs reduces African swine fever virus load in infected pigs and delays virus transmission. Study 1.”, confirmed once again the reliable nature of a model that could be translated to any other laboratory, always keeping the mentioned ratio. Moreover, we consider that this is a more natural model of infection compared to the one in which we inject the virus intramuscularly.

Ethical restrictions (mandatory by our authorities) made impossible to justify duplicating the number of groups and animals used at once, so future experiments will confirm the results observed by using alternative challenge models including intramuscular and intranasal ASFV inoculation with the Georgia07 strain. Despite this additional information will be extremely useful for the field, we believe that our data is solid enough to deserve publication.

Regarding the power calculations to confirm the study design, the reviewer should consider that this study was exploratory to prove the concept, it was not a clinical study of validation. Agreeing that this should be clarify in the manuscript, we have added a new sentence in line 384 of the revised version of the manuscript. 

Finally, the experimental conditions of the study have been added to the new version of the manuscript as following “The rooms contain slatted floor and the environmental conditions for both rooms were set at 22±2°C and relative humidity of 60±5%. The air renewal was established to be 12 times/hour. The feed was provided each morning between 7:30-9:30 am.” Lines: 112-115

Line 231 to 233. Please include the data demonstrating that the virus was Georgia 2007/1 and not the vaccine.

We thank Reviewer #2 for noticing that these results were missing. New Table S4. “Pigs vaccinated with BA71ΔCD2 do not show detectable vaccine virus in blood after Georgia2007/01 challenge” has been added showing the results of the qPCR in blood of BA71DCD2-vaccinated animals after Georgia 2007/01 exposure detecting LacI, only present in the genome of the vaccine virus. The corresponding description of the qPCR method is included in the M&M section lines 154-159.

Figure 5C. Figure seems to be truncated at 140 SFU. Any statistics should be based on the actual values.

We appreciate the comments from reviewer. Following the suggestion, the new version of the manuscript includes a new Fig. 5C without the bars cut from the top.

Figure 6. Please report the actual P values for these comparisons, the variance in the data is so wide that I question whether the statistical differences are biologically significant? Were the data analyzed using repeated measures ANOVA/Mixed Effect Model?

The P values corresponding to Figure 6 (for each cytokine and evaluated day) are shown in the Supplementary Table S7. The statistical analysis for cytokines was conducted by the Epidemiological team at IRTA-CReSA as follow: “A linear mixed-effect model was constructed for each cytokine, with animal and replicate as fixed effects, and treatment and day of study as random effects. In order to contrast both treatments at each time point a post-hoc analysis was performed”. This information has been added to the revised version of the manuscript (lines 192-195)

Line 361. Oura et al. compares pigs with and without CD8 cells, it is difficult to draw the conclusion stated by the authors about elevated CD8 T-cells. Takamatsu 2013 and Goatley 2022 shows elevated CD8 cells in vaccinated animals that are suffering from disease.

We agree with Reviewer #2 that the statement was at least, confusing. It has been changed to: ” Markowska-Daniel and Pejsak [43] reported that the administration of SDPP through both water and feed significantly increased, especially in smaller pigs, the percentage of CD8+ T cells, which are critical for protection against ASFV [10]”.

Round 2

Reviewer 2 Report

The corrections and clarifications that the authors have made have answered all my queries. It's very interesting data and will make an excellent contribution to the field.